# Magnetoresistive-coupled transistor using the Weyl semimetal NbP

Lorenzo Rocchino[1] ✉, Federico Balduini[1], Heinz Schmid [1], Alan Molinari[1], Mathieu Luisier [2], Vicky Süß[3], Claudia Felser [3], Bernd Gotsmann [1] & Cezar B. Zota[1]

Semiconductor transistors operate by modulating the charge carrier concentration of a channel material through an electric field coupled by a capacitor. This mechanism is constrained by the fundamental transport physics and material properties of such devices—attenuation of the electric field, and limited mobility and charge carrier density in semiconductor channels. In this work, we demonstrate a new type of transistor that operates through a different mechanism. The channel material is a Weyl semimetal, NbP, whose resistivity is modulated via a magnetic field generated by an integrated superconductor. Due to the exceptionally large electron mobility of this material, which reaches over 1,000,000 $cm^2/Vs$, and the strong magnetoresistive coupling, the transistor can generate significant transconductance amplification at nanowatt levels of power. This type of device can enable new low-power amplifiers, suitable for qubit readout operation in quantum computers.

The recently discovered topological materials have drawn tremendous research attention due to their unique electronic properties that could be exploited for improved or new kinds of electronic devices[1,2]. Among these materials, Weyl semimetals (WSMs) are of particular interest due, in part, to the presence of relativistic pockets hosting Weyl fermion quasiparticles[3,4]. These quasiparticles are located at physically separated linear band crossing points, i.e., Weyl nodes, which behave as a source or sink of Berry curvature in momentum space[2]. Each Weyl node is associated with a given chirality, leading to unusual transport phenomena such as chiral anomaly and topological protection[5,6]. The linear energy dispersion of these materials translates to a low effective carrier mass and extremely high mobilities[7]. For instance, carrier mobilities of $5 \times 10^6$ $cm^2/Vs$ and $4 \times 10^6$ $cm^2/Vs$ have been reported for the WSMs NbP and WP$_2$, respectively[8]. The high mobility correlates with the observed extreme magnetoresistance (XMR) in these materials, i.e., the strong modulation of the resistivity with respect to an applied magnetic field, which reaches values in the order of $1 \times 10^6$% for NbP and WP$_2$[9,10].

To exploit the high carrier mobility for device functionality, local control of material properties, for instance through an electrostatic gate, is required to realize integrated Weyl electronic devices. However, the quasi-metallic carrier concentration in these materials[8], which for certain materials reaches up to $10^{22}$ $cm^{-3}$, makes electrostatic coupling through electric field-effect gating difficult to implement[11,12]. Nevertheless, the first prototypes of Weyl semimetal-based devices have been proposed and realized, such as novel field-effect transistors[13–19], optoelectronic devices[20], photodetectors[21], and more[1,22].

Many of the unique transport properties of WSMs require applied magnetic fields to be observed, such as chiral anomaly[23,24], extreme magnetoresistance[9,25], anomalous thermoelectric effects[26], etc. Therefore, integration of WSMs with local magnetic field generation to control these properties can enable new functional Weyl devices, such as magnetic-gate transistors. In fact, simulations have shown that magnetic field control of the extreme WSM magnetoresistance can lead to positive high-frequency current and power gain[27]. Moreover, by using magnetic fields instead of electric fields as control quantity, the performance of magnetically-coupled devices is less reliant on having ultra-thin channels, such as is the case for semiconductor electric field devices.

[1]IBM Research Europe—Zürich, Saümerstrasse 4, 8803 Rüschlikon, Switzerland. [2]ETH Zurich, Rämistrasse 101, 8092 Zürich, Switzerland. [3]Max Planck Institute for Chemical Physics of Solids, Nöthnitzer Straße 40, 01187 Dresden, Germany. ✉e-mail: lorenzo.rocchino@zurich.ibm.com

However, this kind of Weyl transistor has not yet been demonstrated. Nanofabrication of such device structures is challenging due to the lack of straightforward and reliable WSM thin-film epitaxy[28]. While there are reports of thin-film epitaxy, the majority of research so far has been done on off-chip bulk single crystals[29,30]. As an alternative, focused ion beam (FIB) manipulation has been successfully used to structure WSM single crystals into complex test structures and can provide a path for realizing new device concepts at sub-micron dimensions[31,32].

Due to the exceptionally high mobility of WSMs, an advanced Weyl transistor could achieve power gain at DC power consumption around a few tens of nanowatts, significantly lower than what is possible with alternative transistor technologies, as was previously demonstrated through simulations[27]. Comparable low-noise amplifiers based on state-of-the-art InP/InAs high-electron-mobility transistors (HEMTs) operate with DC power above 100 μW[33]. Such transistors are part of cryogenic low-noise amplifiers (LNAs) at the qubit readout stage of quantum computers[33]. Qubit readout stages today comprise at least a parametric amplifier at 10 mK, and a cryogenic LNA at 4 K, providing most of the low-temperature gain[33]. The scalability of readout stages due to these components presents a challenge to the development of future quantum computers with large number of qubits. A single compact amplifier stage at 10 mK providing sufficient gain (>40 dB) at extremely low-power dissipation would constitute an ideal solution.

In this work, we report the experimental realization of a magnetoresistive-coupled Weyl transistor. We implement electrical control of a Weyl semimetal NbP microcrystal through local magnetic field generation via a superconducting gate and show that this type of transistor operates with extremely low-power dissipation and promising electrical characteristics due to the high mobility and magnetic field response of these materials.

## Results

### Device concept

The realized Weyl transistor exploits the magnetoresistive coupling between a metal electrode, that acts as the control gate, and the active channel material, the Weyl semimetal NbP. Figure 1a shows a false-colored scanning electron micrograph of the fabricated device (see "Methods" for details on the device fabrication). The control gate is made of superconductive NbN and is electrically insulated from the NbP crystallite by a 20 nm layer of $SiO_2$. The insulator prevents gate leakage currents while allowing the magnetic field to propagate into the WSM. A schematic image of the transverse cross-section of the

device is reported in Fig. 1b. The length, width, and thickness of the NbP are 9, 6, and 1 μm, respectively. The length refers to the distance between the contacts 2–6 and 3–7. Further device dimensions are reported in the Supplementary Materials (Supplementary Table S1). In this kind of transistor, the width also represents the effective gate length. The dimensions and the overall design of the WSM crystallite are chosen to favor uniform current injection parallel to the gate line. In this way, the current and the applied magnetic field are perpendicular, which is necessary in order to maximize the transverse MR[34]. In particular, the current in the NbP crystallite flows along the $x$ direction, which is parallel to the $b$-axis of NbP. The magnetic field is applied along the $y$ direction, parallel to the $a$-axis.

The working mechanism of the transistor action relies on the WSM resistivity modulation induced by the magnetic field generated by the supercurrent flowing in the gate electrode. The device operates as a transistor in the sense that it is a tunable resistor. Considering this, the target operation for the developed device differs substantially from the Weyl-based devices that have been reported in the literature, where topological phase change[16], electrostatic gating[13,14,17,18] or spin-dependent properties[15,19] are primarily exploited to perform the switching operation.

A schematic representation of the principle of magnetoresistive coupling is shown in Fig. 1c. An electric current is induced in the superconductive gate by a DC bias, representing the input signal, applied to the gate electrode. The current generates an approximately uniform magnetic field applied perpendicularly to the current flow in the NbP crystallite, thereby modulating its resistivity. In order to characterize the device, electrodes 2, 3, 6, and 7 were added to enable four-probe measurements of the NbP channel resistance. The current is applied to the external contacts 1 and 5 and collected from contacts 4 and 8, while the voltage is collected by the electrodes located at the contacts 2 and 3 (or, alternatively 6, 7).

### Magnetotransport properties of the NbP microcrystal

The Weyl transistor operation relies on the extreme magnetotransport properties of the NbP crystallite, which are first studied in the presence of a uniform, externally generated magnetic field. Typically in these materials, MR exhibits either a quadratic[35–37] or a linear[38,39] dependence on the magnetic field. In contrast to normal metals, the behavior appears to be non-saturating and can reach up to a few million percent at 9 T, 2 K[40]. Certain WSMs, including NbP, have a parabolic to linear transition at relatively low magnetic field strengths[9,40–43].

In Fig. 2a, we report the magnetoresistive properties of our patterned NbP crystallite at −9 T to 9 T external magnetic field strengths,

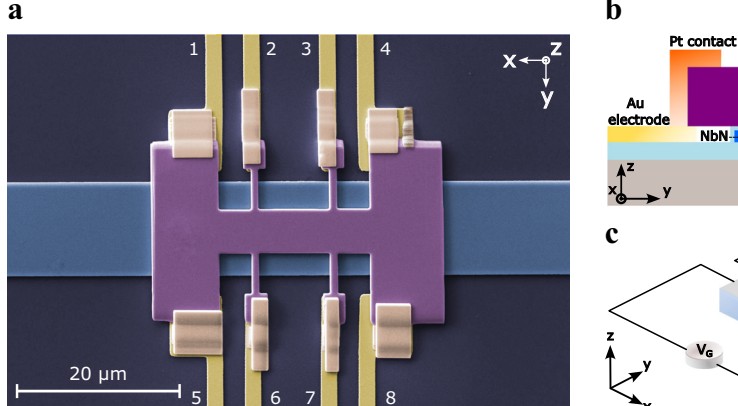

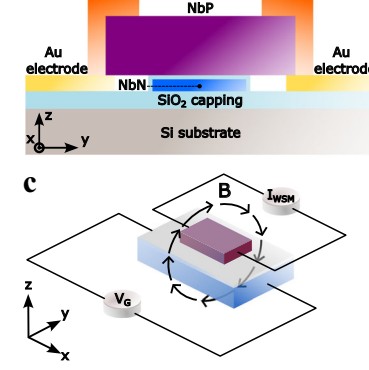

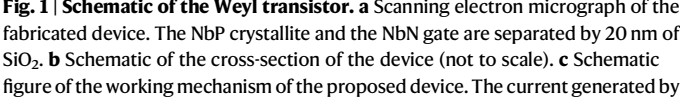

**Fig. 1 | Schematic of the Weyl transistor. a** Scanning electron micrograph of the fabricated device. The NbP crystallite and the NbN gate are separated by 20 nm of $SiO_2$. **b** Schematic of the cross-section of the device (not to scale). **c** Schematic figure of the working mechanism of the proposed device. The current generated by the gate voltage $V_G$ induces a magnetic flux density B which is applied to the NbP, thus modulating its resistance. The resistance variation is measured at constant current $I_{WSM}$ by detecting the voltage variation between the inner contacts (2, 3 or 6, 7).

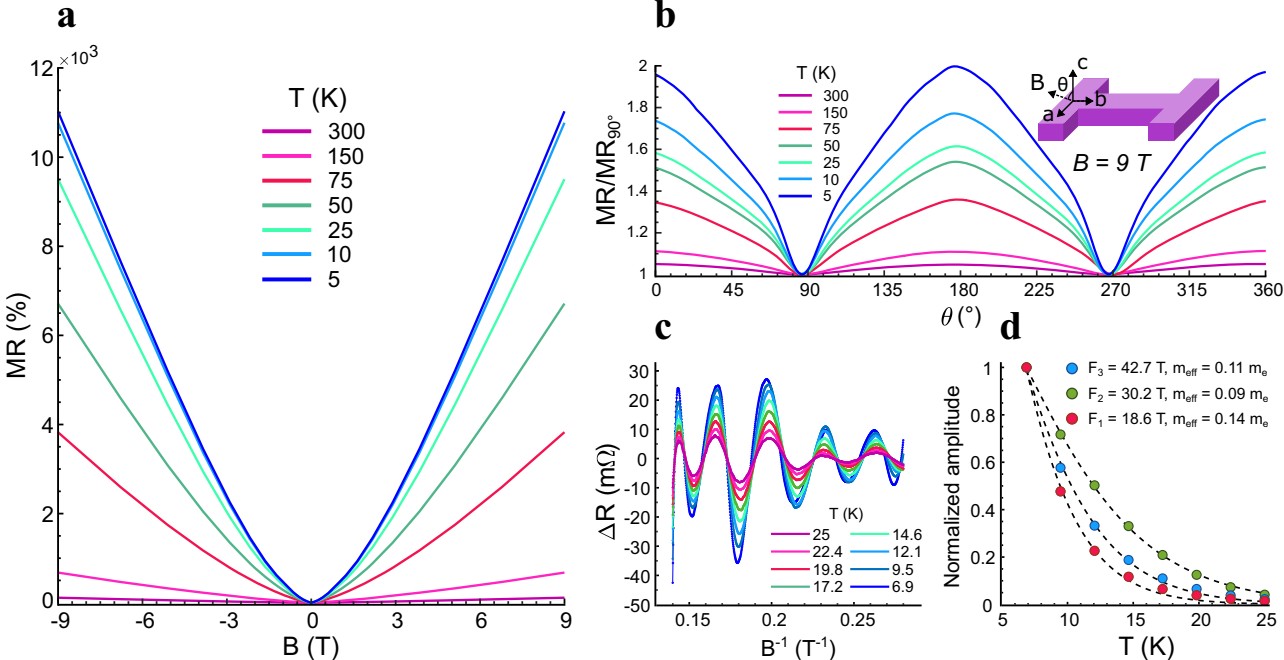

**Fig. 2 | Characterization of the NbP crystallite in an external magnetic field.**
**a** Measured transverse magnetoresistance of the NbP crystallite for temperatures T ranging from 5 to 300 K. **b** Angular dependence of the MR. Note that the magnetic field is always perpendicular to the current flow. **c** Quantum oscillations (Shubnikov-de Haas) extracted from (**a**) by subtracting the non-oscillatory component, which is evaluated with a 4th order polynomial fit, and plotted with respect to 1/B. **d** Calculation of the effective masses for the three main frequencies, obtained by fitting the thermal damping factor (Eq. (1)).

for temperatures between 5 K and 300 K. At 5 K, the MR reaches a maximum of 11,000%. Below 5 K, we can no longer accurately measure the magnetoresistive properties of this NbP crystal, due to a superconductive transition of a FIB-induced Nb-rich surface layer that lowers the zero-field resistivity[44] (Supplementary Fig. S1). In fact, Nb and P have significantly different sublimation points, which causes P to be sputtered when exposed to the ion beam, thus leaving a Nb-rich surface layer. The latter is responsible for the observed superconductivity[44]. The measured MR at ± 9 T is consistent with literature values for FIB-processed NbP crystallites[9,34], as the FIB manipulation lowers the MR by approximately one order of magnitude with respect to bulk crystals[31,45].

Due to the nonuniform local magnetic field generation, the angular dependence of the transverse MR will influence device operation. In Fig. 2b, we report the MR behavior at B = 9 T as a function of the angle of application $\theta$. The magnetic field is rotated by 360° in the plane defined by the a and c lattice vectors, while still being applied perpendicular to the current flow (b-axis). The measured behavior shows a sinusoidal angular dependence which is predicted for high-mobility semimetals[46]. This, together with the almost perfect symmetry of the transverse MR measurement reported in Fig. 2a, suggests a negligible current jetting effect[46,47]. In our device, most of the field is applied parallel to the a-axis due to process-related reasons. While convenient from a fabrication point of view, this configuration is not optimal to maximize the MR response. However, it is important to highlight how, despite a marked anisotropy, the transverse MR response in NbP is within the same order of magnitude in the whole angular range, meaning that the impact of nonuniform distribution of the magnetic field generated by the superconductive gate is mitigated. Other WSMs, such as MoTe$_2$ show stronger anisotropy[48]. Moreover, the geometry of the device has been optimized for relatively uniform magnetic field generation, up to 93% uniformity according to simulations (Supplementary Fig. S2a).

In order to determine charge carriers' mobility, we report in Fig. 2c and d the quantum oscillations extracted from the MR in Fig. 2a

and the amplitude of the oscillations as a function of the temperature, respectively[49,50]. From this, it is possible to calculate the effective mass of the carriers associated to the detected frequencies. The relevant physical quantities calculated from quantum oscillations analysis are reported in the Supplementary Materials (Supplementary Table S2). Quantum oscillations are isolated by subtracting a 4th order polynomial fit from the MR, in a temperature range between 6.9 K and 25 K (i.e., above the superconductive transition of the Nb surface layer) and then plotted as a function of 1/B. The effective masses of the charge carriers in the main pockets are obtained by fitting the thermal damping factor, $R_T$, of the measured oscillation amplitude versus temperature (Eq. (1))[50,51].

$$R_T = \frac{2\pi^2 \, {}^{k_B T}/\hbar\omega_c}{\sinh\left(2\pi^2 \, {}^{k_B T}/\hbar\omega_c\right)} \qquad (1)$$

Here, $k_B$ is the Boltzmann's constant, $T$ is the temperature, and $\hbar$ is the reduced Plank's constant. The effective mass $m_{eff}$ is determined from the cyclotron frequency ($\omega_c = eB/m_{eff}$, where $e$ is the electron charge) by inserting in the above equation the average value of the magnetic field strength over which the oscillations are sampled. The frequencies associated with the main electron and hole pockets obtained from our analysis are consistent with the values reported in literature[9,52,53] ($F_1 = 18.6$ T, $F_2 = 30.1$ T, $F_3 = 42.7$ T). In the main electron pocket, corresponding to the frequency $F_2$[9], carriers have an effective mass of about 0.09 $m_e$ (free electron mass) and are therefore highly susceptible to external perturbations such as a magnetic field. Moreover, the relatively long mean free path of tens of micrometers (Supplementary Table S2) suggests that electron transport at cryogenic temperatures is ballistic along the entire WSM channel of the device in Fig. 1a. Combining the effective mass and the charge carrier density, we evaluate the electron mobility, which reaches values in the order of 1,000,000 cm$^2$/Vs. It is worth noting that these very high values directly relate to the extreme MR exhibited by WSMs[10,38,54], and

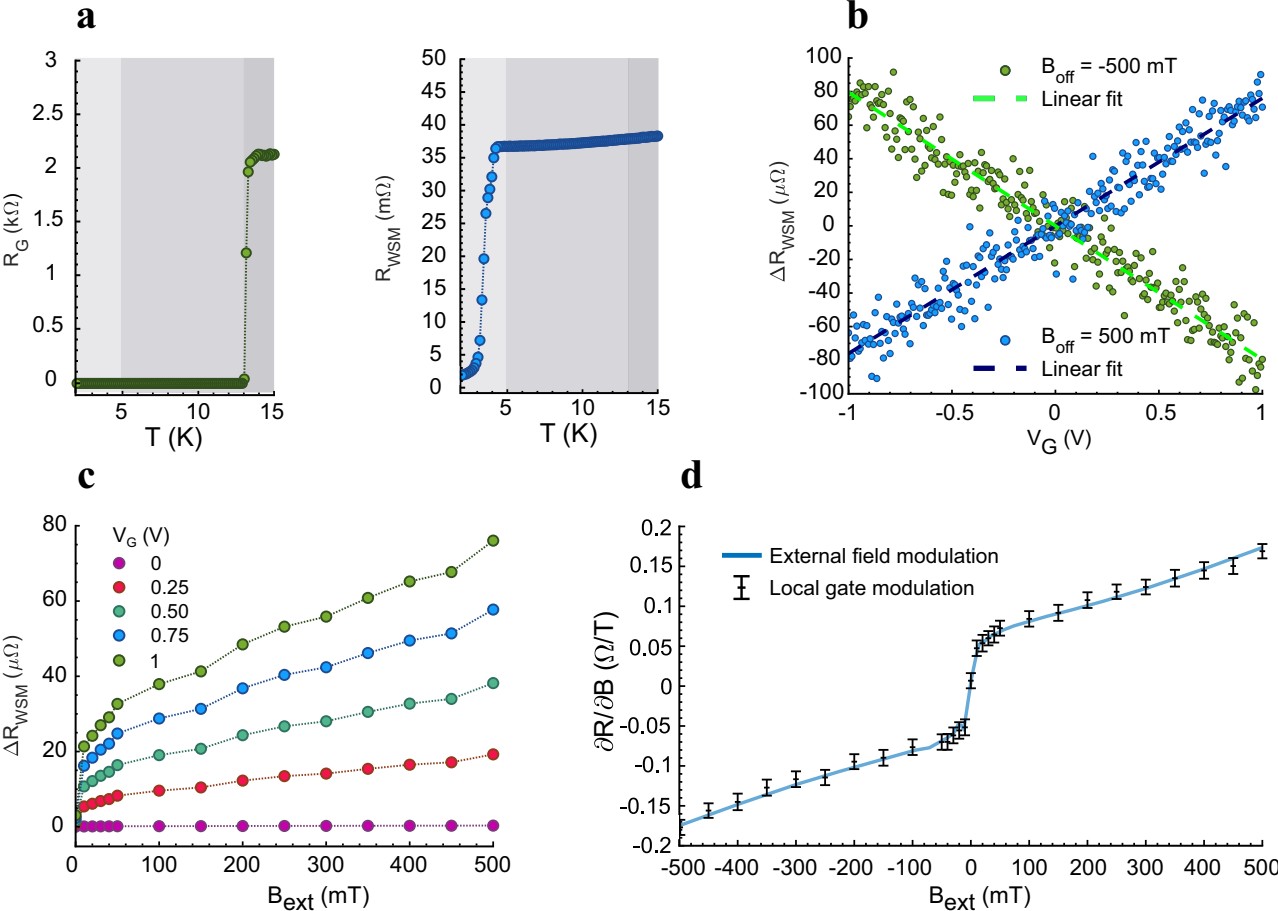

**Fig. 3 | Superconductive characterization and magnetic coupling mechanism.**
**a** Left: Measured resistance of the NbN gate as a function of temperature. Sharp superconductive transition observed between 13.1 K and 13.4 K. Right: Measured resistance of the NbP crystallite as a function of temperature. The observed superconductivity is induced by the FIB procedure, which generates a Nb-rich surface layer. The transition is broad and occurs between 2.1 K and 4.3 K. Left and right: The light-gray area highlights the range in which both NbP and NbN are superconductive, the dark-gray one where none of them is, and the gray area in between highlights the operating window in which only the gate is

superconductive. **b** Measured data points for the gate-induced resistance modulation, taken at opposite field offset. The two slopes are similar (apart from the sign) and in accordance with the local slope of the MR taken with external field. **c** Offset versus resistance variation: introducing an offset forces the modulation to occur around a non-zero slope portion of the MR, thus increasing its effectiveness. **d** Superposition of the external and local field modulation: the effective generated field strength is 0.44 mT. The error bars represent the standard deviation of the individual data points with respect to the linear regression (as shown in (**b**)).

therefore also to the effectiveness of the magnetoresistive coupling in the transistor device.

## Weyl transistor properties

We proceed to characterize the Weyl transistor shown in Fig. 1a. The superconductive NbN gate is implemented to eliminate the contribution of self-heating, caused by the power dissipated in the gate electrode, to the resistivity modulation. The superconductive transition of the NbN gate is reported in Fig. 3a, left. The transition occurs in the temperature range of 13.1 to 13.4 K. This value is in accordance with the $T_C$ reported in literature[55] and confirm the good quality of the deposited film. As previously described, due to the influence of the FIB fabrication process, the NbP crystallite also exhibits a superconductive transition, shown in Fig. 3a, right. The transition happens between 2.1 and 4.3 K and is less sharp, as expected, given the spurious state of the Nb surface layer. Therefore, in this temperature range, the observed MR does not represent a true property of the NbP crystallite. For this reason, the working temperature for the Weyl transistor in the following measurements is set to 5 K, where the NbN gate, but not the NbP, is superconductive.

We confirm the transistor operation via the superconducting gate by combining the local generated magnetic field with an external offset field generated by the measurement setup. The external offset field can be exploited to lift the operation point towards a higher dR/dB (Supplementary Fig. S3). This is beneficial in order to maximize the magnetoresistive coupling between the gate and the channel. Furthermore, by sweeping the external magnetic field while measuring the local gate modulation, we can confidently attribute a magnetic origin to the measured signal. The resistivity modulation is first measured at −500 mT and 500 mT external field offsets. The results are reported in Fig. 3b. The resistance modulation values at the two different offsets have the same magnitude (roughly 150 μΩ at gate voltages from −1 V to 1 V), but opposite slopes. Since the MR modulation is expected to occur along the two opposite branches of the parabola, this observation is a strong indication that the signal is due to magnetic coupling, rather than parasitic effects such as heating and leakage. If we normalize the resistance variation to the zero-field value, we obtain a modulation effectiveness of 0.45%/V.

In Fig. 3c, a study is reported of the variation in resistance modulation, at different gate voltages (nominally, from 0 up to 1 V), while

sweeping the external offset field between 0 T and 0.5 T. The effectiveness of the gate modulation, i.e., dR/dB, is increased for higher field offsets, meaning that the magnetoresistive coupling in the transistor is enhanced. Three paths to device operation without an external field are envisioned: (i) Channel materials with steeper MR response close to 0 T. (ii) Advanced gate geometries based on planar device fabrication, given the availability of epitaxial high-MR thin films. (iii) Magnetic field focusing.

Figure 3d shows the superposition between the first derivative of the MR data (dR/dB), evaluated from the same dataset as in Fig. 2a, and the local variation in resistance at a given bias point ($\Delta R/\Delta B$). The two datasets are compared by matching the resistance values measured when the gate voltage is zero. This approach allows to transduce the current flowing in the gate into a magnetic field strength value. The result obtained by matching the resistances is that the produced field strength, sensed by the WSM, is equal to 0.44 mT. This value is obtained by applying a nominal gate voltage of 1 V, which generates a supercurrent of 13.5 mA flowing in the gate. According to a numerical simulation performed with ANSYS Maxwell 3D, by solving Maxwell's equations in a transverse cross-section of the active area of the device, the average magnitude of the magnetic field strength of the NbP crystallite is 0.48 mT (Supplementary Fig. S2a). This result is close to the measured value of 0.44 mT, strengthening the conclusion that the device indeed operates through a magnetic coupling.

A possible source of deterioration of the magnetic signal is self-heating of the device. The superconductive gate does not self-heat, but the NbP crystallite could at large biases. To study this, Fig. 4a reports the resistivity variation of the WSM at different gate biases, for an effective voltage drop on the NbP crystallite up to 80 μV, with a field strength offset of 500 mT. The resistivity increase with increasing NbP voltage indicates that the device starts to marginally heat up at around 50 μV. To maximize the resolution, the previous measurements are taken at 80 μV. Significant self-heating of several Kelvin would cause a reduction of MR (Fig. 2a) and result in an underestimation of the sensed local magnetic field, since the MR would be lower than the assumed one. However, this is not the case, given the good accordance between the measured results and the simulations, in which self-heating is not taken into account.

From the resistivity versus voltage characteristic, it is possible to extract the transconductance $g_m$ of the device, which is an important figure-of-merit for the high-frequency amplifying properties of the transistor. Starting from the general definition of transconductance, its behavior as a function of the applied WSM bias is evaluated

(Eq. (2)). A full derivation is provided in the Supplementary Materials.

$$g_m \overset{\text{def}}{=} \frac{\partial I_{out}}{\partial V_{in}} = \frac{\partial I_{WSM}}{\partial V_G} = -\frac{V_{WSM}}{K_{WSM}} \frac{K_G}{(1+K_G V_G)^2} \qquad (2)$$

Here, $V_{WSM}$ is the voltage applied to the NbP crystallite and $V_G$ is the gate voltage. $K_{WSM}$ and $K_G$ are constant values related to the geometrical and electrical properties of the WSM and the gate, respectively. $K_G$ also depends on the slope of the MR curve and, therefore, on the applied external field offset. Given the theoretically zero resistance of the gate, the value of the extrinsic $g_m$ depends, in practice, on the quality of the contacts to the NbN gate. The results of the $g_m$ measurements are reported in Fig. 4b. The reported values of $g_m$ are obtained by numerically computing the derivative of the output signal with respect to the input signal. We note that the measured trend is linear with respect to the voltage drop $V_{WSM}$, in agreement with Eq. (2). The corresponding current through the WSM reaches ~800 μA at 80 μV, leading to DC power dissipation of ~70 nW. The developed device operates close to a HEMT device, which in a low-noise amplifier is always turned on and consumes a constant power which is generally dominated by the static power. Therefore, the power dissipation we report is to be intended as static power. To compare with standard transistors, such as HEMTs and Si metal-oxide-semiconductor field-effect transistors (MOSFET), we normalize $g_m$ to the narrowest section of the NbP crystallite, 6 μm. This section is comparable to the gate width of a field-effect transistor, since it can be extended in order to increase the current without impacting device performance. We thus obtain normalized transconductance of 22 mS/μm at 80 μV applied WSM bias. However, as we noted, the high $g_m$ in this Weyl transistor is due not only to the extremely low resistivity of the channel, but also to the signal transfer from the superconducting gate to the WSM. Therefore, the high-frequency properties, such as the power gain, are different from standard transistors and a direct comparison with respect to high-frequency properties cannot be made strictly based on DC characteristics[27]. Nevertheless, the combination of extremely high channel mobility and strong magnetoresistive coupling is predicted to enable a level of energy efficiency that is not possible with semiconductor transistor technologies[27].

As a direction for future improvements, device designs based on scaled Weyl semimetal thin films[29] that can generate stronger local magnetic fields could be explored. Preliminary results on semimetal thin-film growth exist, such as for Cd$_3$As$_2$[56–58], HfTe$_2$[59], PtTe$_2$[60], NbP[29], TaP[29], and Na$_3$Bi[61], among others. In addition, the use of new channel

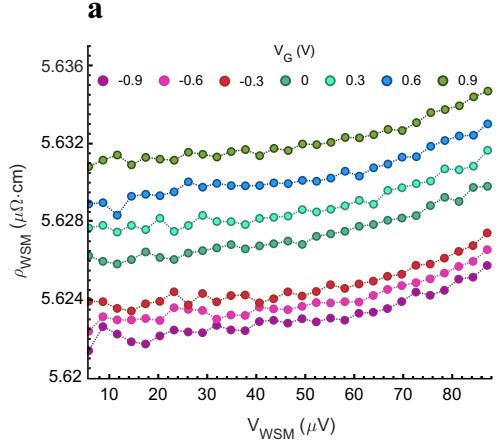
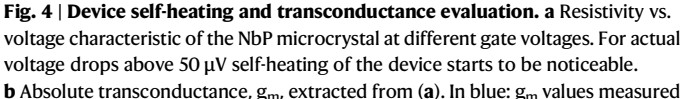
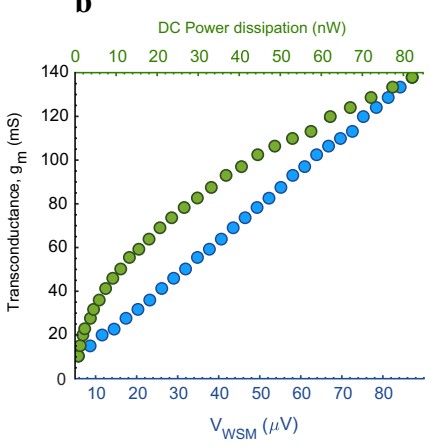

**Fig. 4 | Device self-heating and transconductance evaluation. a** Resistivity vs. voltage characteristic of the NbP microcrystal at different gate voltages. For actual voltage drops above 50 μV self-heating of the device starts to be noticeable. **b** Absolute transconductance, $g_m$, extracted from (**a**). In blue: $g_m$ values measured

as a function of the voltage drop on the WSM. The linear trend is in agreement with Eq. (2). In green: measured $g_m$ as a function of the power dissipated in the WSM. The transconductance is evaluated with respect to the actual voltage drop on the gate contacts (~8 μV at a nominal value of 1 V).

materials with strong magnetic field response close to zero field[62], or with enhanced MR response[63], can enable operation without an external magnetic offset field. transport properties of such materials.

## Discussion

In this work, we have demonstrated a magnetic field-effect transistor based on the Weyl semimetal NbP. The transistor action is enabled by the extreme magnetoresistance of this material, which is modulated through the local magnetic field generated by an integrated superconductor. The device utilizes a FIB-manipulated NbP microcrystal exhibiting carrier mobility greater than 1,000,000 $cm^2$/Vs, and magnetoresistance above 10,000% at 5 K and 9 T. These properties enabled the Weyl transistor to operate with extremely high transconductance gain at nanowatt levels of power dissipation, indicating the potential for improvements over standard cryogenic amplifier technologies. We foresee qubit readout signal amplification as a highly attractive area of application for this type of device due to the increasing need for low-power cryogenic amplifiers in advanced quantum computers. The results indicate a promising path forward for integrated Weyl semimetal electronics that can leverage the often extreme transport properties of such materials.

## Methods

### Sample growth

The NbP crystal was grown in the same batch as the ones characterized in previous works[32,64]. Polycrystalline NbP powder was synthesized by reacting Nb powder (99.9% purity) and red phosphorous pieces (99.999% purity) in an evacuated silica tube for 48 h at 800 °C. Single NbP crystals were then grown from this powder through a chemical transport reaction using iodine as the transport agent, with the source and sink set at 950 °C and 850 °C, respectively.

### Device fabrication

The gate process consisted of the following steps. First, 200 nm of superconductive NbN was deposited by means of physical sputtering deposition. The NbN layer was subsequently covered with a $SiO_2$ hard mask (400 nm) obtained by means of plasma-enhanced chemical vapor deposition (PECVD). The hard mask was patterned with a direct write laser lithography (Heidelberg DWL 2000) and then opened with a reactive ion etching (RIE) process. Finally, the NbN layer was etched into the desired shape with inductively coupled plasma reactive ion etching (chlorine-based), and the $SiO_2$ hard mask was removed with HF. Prior to the crystallite deposition, 20 nm of PECVD $SiO_2$ was deposited, to provide electrical insulation. Au contact pads (210 nm Au + 10 nm Ti, for adhesion) were evaporated and patterned with a lift-off method. Lastly, from the NbP bulk sample, a 30 μm × 20 μm × 1 μm large crystallite was cut and then patterned with a dual beam focused ion beam system (FIB) of the type FEI Helios 600i using 30 keV Ga+ ions. The crystallite was then transferred onto the gate structure by using an Omni-probe integrated in the FIB system.

### Electrical transport measurements

The measurements were carried out using a Quantum Design PPMS system, a cryostat able to cool the sample down to 1.7 K and to generate a magnetic field strength up to 9 T. The PPMS system was externally connected to two lock-in amplifiers for the electrical measurements (Zurich Instruments MFLI 500 kHz/5 MHz). The measurements were taken applying a constant current at -113 Hz. The reported values of current and voltage are root mean square (RMS) values.

### Data analysis and simulations

The simulations have been carried out first within the MATLAB environment, implementing a self-made code for the field strength evaluation, which solves numerically a generalized Biot–Savart equation. For further validation of the results, ANSYS Maxwell 3D has been used,

solving Maxwell's equation in a transverse cross-section of the device. Both MATLAB and a Python environment were used for the data analysis.

## Data availability

The data generated in this study is available on Zenodo under the accession code https://doi.org/10.5281/zenodo.10276238.

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

## Acknowledgements

We wish to acknowledge the support of the Cleanroom Operations Team of the Binnng and Rohrer Nanotechnology Center (BRNC). We thank Daniele Caimi for the deposition of the NbN layer. L.R. and C.Z. acknowledge the SNSF Ambizione programme (no. 193636). C.Z., H.S., A.M., V.S., C.F., and B.G. acknowledge the FET open project no. 829044 (SCHINES). F.B. and B.G. acknowledge the SNSF open project HYDRONICS, under the Sinergia grant (no. 189924). A.M. acknowledges funding support from the European Union's Horizon 2020 research and innovation program under the Marie Sklodowska-Curie grant agreement no. 898113 (InNaTo).

## Author contributions

L.R., C.Z., and B.G. conceived and developed the idea of the Weyl transistor. C.F. and V.S. grew the NbP crystal. L.R. fabricated and characterized the device, with the help of H.S. and F.B. (FIB operation). L.R. and F.B. collaborated in the measurement phase and for the data analysis. L.R., C.Z., B.G., A.M., F.B., H.S., and M.L. discussed about the interpretation of the data. L.R. wrote the manuscript with input from all authors.

## Competing interests

The authors declare no competing interests.
