## [Peer Review File · Nature Communications]

Magnetoresistive-coupled transistor using the Weyl semimetal NbPREVIEWER COMMENTS

Reviewer #1 (Remarks to the Author):

Lorenzo Rocchino et al. reported the novel transistor based on the magnetoresistive-coupling mechanism with Weyl semimetal NbP. In this transistor, electron mobility is modulated via a magnetic field, which could be generated by an integrated superconductor. Overall, this work suggests a new-mechanism transistor with Weyl materials, with relatively low power consumption (nanowatt levels). The reviewer will recommend publication in Nature Communications if the authors can address the following issues.

1. The NbP family (TaAs, TaP, TaP and NbP) is a well-known high-mobility system with high magnetoresistance. In this work, the authors have used a superconductor to induce the local magnetic field. Is this kind of magnetic field strong enough to affect its mobility and magnetoresistance? Maybe the performance is not good enough for the practical application. To convince readers, the authors could give some specific applications, in which the proposed transistors can overwhelm conventional devices.

2. The title and abstract highlight the application of "transistor" with Weyl materials. However, the main text doesn't present enough discussion of transistor features. Recent works on Weyl materials-based transistors could be discussed together, such as:

2.1 "Electric control of Fermi arc spin transport in individual topological semimetal nanowires."

Physical Review Letters 124.11 (2020): 116802. This work concludes that "The gate control of Fermi arc spin-polarized transport should be promising for the topological field effect transistors".

2.2 "Topological phase change transistors based on tellurium Weyl semiconductor." Science Advances 8.23 (2022): eabn3837. This kind of transistor is based on the gate-tunable topological phase change in Weyl materials.

2.3 "Room-temperature valley transistors for low-power neuromorphic computing." Nature Communications 13.1 (2022): 7758.

2.4 "Chirality-induced intrinsic charge rectification in a tellurium-based field-effect transistor." Physical Review B 106.22 (2022): L220403.

3. One key parameter of the transistor is the ON/OFF ratio. Fig. 3c and Fig. 4a have presented the effects of gating voltages on the transistors, which may be used for calculating the ON/OFF ratio.

4. The authors claim "the transistor can operate at nanowatt levels of power". Generally, there are both static and dynamic power consumption for transistors. Please calculate them respectively and compare them with reported works.

5. The authors claim in the conclusion that "The device utilizes a FIB-manipulated NbP micro-crystal exhibiting carrier mobility greater than 1,000,000 cm²/Vs, and magnetoresistance above 10,000% at 5 K and 9 T". The testing condition is harsh. 9 T is too high for the practical application. In lines 234-236, the authors propose some possible solutions but the description is not clear. Please elaborate the possible mechanism and material for testing under more accessible conditions.

6. Give out the thickness of different parts (e.g., Pt contact, NbP and NbN in the transistor). It will be more readable if presenting the cross-section TEM image of the device.

7. Minor issues: in the references, many numbers should be in the form of "subscript", such as references [10] and [22]. Please check the references carefully.

Reviewer #2 (Remarks to the Author):

This paper describes the new concept of the cryogenic transistor using magneto-resistive coupling between a superconductor and Weyl semimetal. Basically, I enjoyed reading this manuscript because the concept of the device is very new to me and an interesting idea to be implemented for cryogenic applications. However, I strongly recommend authors consider the following comments.

1. The device fabrication process seems to be tricky, therefore, I am curious about the repeatability of the device's performance.
2. Readers would want to know about the material's properties. Could authors include the material characterization data such as XRD, TEM, XPS, etc.?
3. Since the mean free path of an electron in NbP is several tens of μm , transport characteristics of this film will experience significant thickness dependence. This potentially limits the actual use of this type of transistor. Could authors comment on this?
4. In a similar line, device scalability should be discussed. As the authors described in the paper, the device surface will not have uniform NbP stoichiometry, limiting the device scaling. Furthermore, the strong angular dependence of MR will be a critical hurdle for scaling.
5. DC feasibility is shown in this paper. However, AC/RF performance has not been discovered yet, whereas this is likely more important for cryogenic applications. Adding these kinds of data will significantly strengthen the impact of this paper.
6. The use of external field offset is not very straightforward to me. It can be one of the performance boosters, but this imposes unfair comparison with other technology.
7. Since this device uses the magnetic field, one of my concerns would be the interference between neighboring devices. Could the author comment on this?

Reviewer #3 (Remarks to the Author):

Review of Magneto-resistive-coupled transistor using the Weyl semimetal NbP

The manuscript presents fabrication and analysis of a Weyl Semimetal based transistor, based on magneto-resistive coupling. Manuscript is well written, easy to read and with high quality figures. The device concept is highly interesting and relevant. The paper looks into how one can utilize the high mobility, conductance and magneto-resistance of a Weyl semimetal for a useful application, towards quantum technology. The FIB device fabrication, although simple, is also of interest for fast evaluation of novel materials.

The data analysis is mainly sound, and convincing evidence is presented that the device is operating as suggested through magneto-resistive coupling. As such, I find that the manuscript could be of interest for nature communications.

However, there are some shortcomings in the manuscript which would need to be addressed for the publication to be fit for nature communications.

1. The device is called a transistor, but seems in reality to operate more as a slightly tunable resistor. Does the device give any voltage and/or current gain (which should be included in the manuscript)? If not, one should reconsider the title. It is somewhat problematic if both values are very small as compared to unity.
2. Qubit amplification readout is suggested. Is this feasible for a device with extremely small input and output impedance?
3. Is it claimed that the device can uniquely operate at nanowatts level, making it unique. This is a bit misleading – more FETs can indeed be operated at nanowatts levels (i.e. subthreshold operation), but with low gain. The unique point here is the large g_m at low V_{ds} .
4. A very impressive g_m is presented, really highlighting the unique possibility of the device. However, given that the resistivity is modulated by the gate current and not the applied voltage,

g_m would then also depend on the resistances used to convert the applied gate voltage into a current. This should be verified.

5. Fig 4a seems to be the only shown data with ρ vs V_g , but does not seem to follow the assumptions leading to eq 2. It could be an idea to include better data with ρ , R or I vs V_g/I_g to highlight the device operation.

RESPONSE TO REFEREES

Reviewer #1 (Remarks to the Author):

1. *The NbP family (TaAs, TaP, TaP and NbP) is a well-known high-mobility system with high magnetoresistance. In this work, the authors have used a superconductor to induce the local magnetic field. Is this kind of magnetic field strong enough to affect its mobility and magnetoresistance? Maybe the performance is not good enough for the practical application. To convince readers, the authors could give some specific applications, in which the proposed transistors can overwhelm conventional devices.*

Reply: The intrinsic mobility (as opposed to effective mobility) should be independent of the magnetic field, rather it is a function of the temperature (abstract has been changed to avoid confusion, line 15). MR depends on B through the mean free path/cyclotron radius. Based on the work reported in our previous simulation work¹, we believe that the performance could be strong enough given future epitaxial layer grown with high quality/new materials exhibiting ultra-high MR². The simulations were based on more ideal material properties and nanofabricated structures, but also included high-frequency properties at frequency ranges relevant to qubit readout. The results showed that useful amounts of gain could be achieved at extremely low power levels, which is the direction that our present experimental work also points in. We report how this type of device is feasible for cryogenic quantum applications in the text (lines 47-48 and 59-70).

Changes: Added clarifications accordingly at lines 61-62. Changed a sentence in the abstract at line 15.

2. *The title and abstract highlight the application of “transistor” with Weyl materials. However, the main text doesn’t present enough discussion of transistor features. Recent works on Weyl materials-based transistors could be discussed together, such as:*
 - a. *“Electric control of Fermi arc spin transport in individual topological semimetal nanowires.” Physical Review Letters 124.11 (2020): 116802. This work concludes that “The gate control of Fermi arc spin-polarized transport should be promising for the topological field effect transistors”.*
 - b. *“Topological phase change transistors based on tellurium Weyl semiconductor.” Science Advances 8.23 (2022): eabn3837. This kind of transistor is based on the gate-tunable topological phase change in Weyl materials.*
 - c. *“Room-temperature valley transistors for low-power neuromorphic computing.” Nature Communications 13.1 (2022): 7758.*
 - d. *“Chirality-induced intrinsic charge rectification in a tellurium-based field-effect transistor.” Physical Review B 106.22 (2022): L220403.*

Reply: We thank the reviewer for the comment. The present device is not meant to be a logic device – as the reviewer points out, it does not have a proper off-state – but rather a transconductance amplifier. In this sense, the device will function much more similar to a high-electron mobility (HEMT) transistor, which are part e.g. in low-noise amplifiers, than a Si CMOS transistor. HEMTs are typically biased in the on-state and are typically always on. A sentence to make this concept clearer has been added to the manuscript (line 95). We thank the reviewer for the additional relevant

¹ Toniato, A., Gotsmann, B., Lind, E. & Zota, C. B. Weyl Semi-Metal-Based High-Frequency Amplifiers. in 2019 IEEE International Electron Devices Meeting (IEDM) 9.4.1-9.4.4 (2019). doi:10.1109/IEDM19573.2019.8993575.

² Xin, N. et al. Giant magnetoresistance of Dirac plasma in high-mobility graphene. Nature 616, 270–274 (2023).

references. The references listed here are now present in the manuscript as citations, and we have added a comment to better highlight how this work position relates to the others.

Changes: Added clarifications accordingly at lines 95-99.

3. *One key parameter of the transistor is the ON/OFF ratio. Fig. 3c and Fig. 4a have presented the effects of gating voltages on the transistors, which may be used for calculating the ON/OFF ratio.*

Reply: We thank the reviewer for the insightful comment. As the reviewer noted, the device does not have a low-leakage off-state, and thus the ON/OFF ratio will be rather small compared to Si CMOS. But as the device operates as a normally-on high-frequency amplifier rather than a logic switch, we believe that the ON/OFF ratio would not represent a meaningful quantity to add to the main text. We characterized the transistor in terms of transconductance amplification (Eq. 2 and Fig. 4b). Nevertheless, we added a sentence in the text to highlight the percentage of resistivity modulation that we achieved in our device, which basically translates to the ON/OFF ratio.

Changes: Added clarifications accordingly at lines 189-190.

4. *The authors claim, “the transistor can operate at nanowatt levels of power”. Generally, there are both static and dynamic power consumption for transistors. Please calculate them respectively and compare them with reported works.*

Reply: We thank the reviewer for this question. Indeed, a high-frequency amplifier will consume both static and dynamic power. The developed device operates close to a HEMT device, which in a low-noise amplifier is always turned on and consumes a constant power which is generally dominated by the static power, similar to what is the case for HEMT, capacitances are generally very low, and the RF power is low. Therefore, the power dissipation we mention in the manuscript is expected to be strongly dominated by the static power component.

Changes: Added clarifications accordingly at lines 234-236.

5. *The authors claim in the conclusion that “The device utilizes a FIB-manipulated NbP micro-crystal exhibiting carrier mobility greater than 1,000,000 cm²/Vs, and magnetoresistance above 10,000% at 5 K and 9 T”. The testing condition is harsh. 9 T is too high for the practical application. In lines 234-236, the authors propose some possible solutions, but the description is not clear. Please elaborate the possible mechanism and material for testing under more accessible conditions.*

Reply: We thank the review for the question. The 9 T used to measure the MR is a more or less standard value used in literature for characterizing the material exhibiting XMR. It is useful to apply such high magnetic fields to probe the behavior of the MR trend, as this trend can provide information about the transport physics of the semimetal. Therefore, we refer to that value to highlight the high MR exhibited by our NbP crystal (it's implicit that the higher the MR is at 9 T, the higher it will be at the operating point, if the trend is parabolic). In general, it stands that the higher the MR or the field generation, the better the performance, and our magnetic off-set fields during the transistor operation is indeed, as the reviewer suggests, significantly smaller, <0.5 T.

Changes: We changed a sentence and added a reference to further corroborate this statement (line 253).

6. *Give out the thickness of different parts (e.g., Pt contact, NbP and NbN in the transistor). It will be more readable if presenting the cross-section TEM image of the device.*

Reply: The thicknesses are reported in the *Supplementary Materials*, apart from the Pt contacts, which we did not measure precisely but, given the deposition rate of Pt, should be in the order of 1 μm . The missing information is now reported in the *Supplementary*.

Changes: Added the estimated thickness of the Pt contacts in the *Supplementary* (Tab. S1).

7. *Minor issues: in the references, many numbers should be in the form of “subscript”, such as references [10] and [22]. Please check the references carefully.*

Reply: Everything looks alright to me. Perhaps there was an issue with sending/downloading the document.

Reviewer #2 (Remarks to the Author):

1. *The device fabrication process seems to be tricky, therefore, I am curious about the repeatability of the device's performance.*

Reply: We thank the reviewer for this comment. The FIB procedure is the only step which might affect the reproducibility, as we have several substrates with the SC gate electrode that have been tested and showed reliable SC properties. Also, the NbP itself has been measured extensively in our group with similar results (we cite other works with analogous data, lines 122-124). Moreover, in terms of real-life applications, epitaxial growth of the WSM would be required, not only for repeatability but for throughput. However, as the reviewer notes, it is clear that the FIB procedure will induce variations in the exact device performance values, but since other groups have used similar methods extensively to characterize these materials, we believe it is a reliable method for proof-of-concept devices.

Changes: We added a sentence and a reference commenting on the SC properties of the NbN gate (lines 170-171).

2. *Readers would want to know about the material's properties. Could authors include the material characterization data such as XRD, TEM, XPS, etc.?*

Reply: We agree with the reviewer that the readers may find this data important. We now report in the *Methods* section the references for the required data (the crystal we used was grown in the same batch as the one reported in those papers).

Changes: Added information as a reference in the *Methods* section (line 269-270).

3. *Since the mean free path of an electron in NbP is several tens of μm , transport characteristics of this film will experience significant thickness dependence. This potentially limits the actual use of this type of transistor. Could authors comment on this?*

Reply: We thank the reviewer for this insightful comment. It is indeed known that scaling down the thickness of a WSM generally worsens their MR and consequently the potential performance of our device. However, thin films might benefit from a more uniform vertical scaling of the magnetic field, partially compensating the previous issue. Moreover, other materials exhibiting huge MR are yet to be discovered. Another important observation is that, compared to a surface-channel transistor, the present device does not rely on surface effects, but rather on bulk conduction. In addition, a magnetic field can be generated without significant attenuation in the z-direction (as opposed to the electric field in a transistor), therefore, the crystal does not as a principle need to be scaled down to nm dimensions.

Changes: We added a reference to further corroborate this statement (line 253).

4. *In a similar line, device scalability should be discussed. As the authors described in the paper, the device surface will not have uniform NbP stoichiometry, limiting the device scaling. Furthermore, the strong angular dependence of MR will be a critical hurdle for scaling.*

R: Concerning stoichiometry and related SC effects, that will not be an issue in an epitaxially grown material. Concerning the scaling of dimensions, not only epitaxy will help, but it has to be noticed how, as we noted in the above answer, by using magnetic fields instead of electric field, the device

performance is less reliant on having ultra-thin channels, such as is the case for semiconductor electric-field devices.

Changes: We added a sentence to better highlight this important aspect in the main text (lines 48-51).

5. *DC feasibility is shown in this paper. However, AC/RF performance has not been discovered yet, whereas this is likely more important for cryogenic applications. Adding these kinds of data will significantly strengthen the impact of this paper.*

Reply: A preliminary analysis on RF performance has been reported here¹. Further analysis on the frequency behavior shows how RF performance is dependent on the signal transfer from the gate to the WSM, which in the present device is affected by the gate resistance (contacts) and the kinetic inductance. In this paper we focus on the first experimental demonstration of the device concept, but we believe the interested reader can obtain the details on potential RF performance from our previous simulation work. RF measurements on the present device is beyond the scope of this paper, as S-parameters will be too small to measure with current devices.

Changes: A section explaining how the gate parameters impact RF performance has been added to the *Supplementary* (lines 34-50).

6. The use of external field offset is not very straightforward to me. It can be one of the performance boosters, but this imposes unfair comparison with other technology.

Reply: The value of 0.44 mT of magnetic field strength generated by our device would not have been enough to provide a full characterization of the device and its working principle. Therefore, we used the external offset as a tool to enhance the magnetic modulation. In an ideal/better version of the device, this would not be necessary, nor we are proposing it as a working condition (even though, if required, on-chip solutions to generate an offset could in principle be implemented by depositing nanomagnets). The use of the offset field also allows us to show that the observed resistivity modulation is due to the locally generated magnetic field and not parasitic effects, which we believe is a highlight of the paper.

Changes: Added clarifications accordingly at lines 182-183.

7. *Since this device uses the magnetic field, one of my concerns would be the interference between neighbouring devices. Could the author comment on this?*

Reply: We thank the reviewer for this important question. We have now added a second image in the Fig. S2 to highlight how the field decays in the device plane and vertically. Vertical decay is slower: this could be exploited (stacked devices) or prevented adding extra layers to shield the magnetic field (if needed). Likewise, the in-plane scaling, which could affect neighboring devices, could be mitigated depositing SC structures which would screen the magnetic field.

Changes: Improved Fig. S2 with a new image addressing how the field scales outside of the device area.

Reviewer #3 (Remarks to the Author):

1. *The device is called a transistor but seems in reality to operate more as a slightly tuneable resistor. Does the device give any voltage and/or current gain (which should be included in the manuscript)? If not, one should reconsider the title. It is somewhat problematic if both values are very small as compared to unity.*

Reply: We thank the reviewer for this question, as other reviewers have pointed out as well. As we described in a previous answer, the device is not meant to be a logic device but rather a transconductance amplifier. A sentence to make this concept clearer has been added to the manuscript (line 95), specifying how indeed the developed device is a transistor in the sense that it is a gate-tunable resistance. Current and voltage gain are dependent on the signal transfer from the gate to the WSM and will therefore depend on the quality of the contacts (see lines 229-230 and 241-243). We added in the supplementary a section on the frequency behavior which addresses the current gain more in depth.

Changes: Added clarifications accordingly at line 95 (main text). Added a sentence in the *Supplementary* addressing current gain (lines 48-49).

2. *Qubit amplification readout is suggested. Is this feasible for a device with extremely small input and output impedance?*

Reply: There are different readout schemes, for instance charge-based readout in spin qubits, that could be an interesting application of the developed device. Given sufficient amplification, operation similar to an HBT could be envisioned. This would be suitable for “transport” readout of spin qubits, as such amplifiers are already employed to directly read currents from readout-SETs. We believe a detailed discussion on specific applications is beyond the scope of this work, but refer kindly to general discussions on qubit readout such as *Krantz, Philip, et al. "A quantum engineer's guide to superconducting qubits." Applied physics reviews 6.2 (2019).*

3. *Is it claimed that the device can uniquely operate at nanowatts level, making it unique. This is a bit misleading – more FETs can indeed be operated at nanowatts levels (i.e., subthreshold operation), but with low gain. The unique point here is the large gm at low Vds.*

Reply: We thank the reviewer for this keen observation. This is indeed true, the low-power level is only relevant in relationship to gain. The device operation requires them to be always in the ON-state, consuming a constant power which is typically way higher than the one reported in this work. We have added comments in the abstract to clarify this point.

Changes: Changed the abstract and added a sentence commenting about power-dissipation (lines 234-236).

4. *A very impressive gm is presented, really highlighting the unique possibility of the device. However, given that the resistivity is modulated by the gate current and not the applied voltage, gm would then also depend on the resistances used to convert the applied gate voltage into a current. This should be verified.*

Reply: Yes, R_G does play a role (lines 229-230 and 241-243). However, given that the gate is SC, the real component of the impedance will be given by the contacts, while the gate itself will mostly affect the frequency response. More information about the impact of R_G is added in a new section of the *Supplementary*.

Changes: Added a new section in the supplementary addressing signal transfer and frequency behavior (lines 34-50).

5. Fig 4a seems to be the only shown data with ρ vs V_g but does not seem to follow the assumptions leading to eq 2. It could be an idea to include better data with ρ , R or I vs V_g/I_g to highlight the device operation.

Reply: Figure 4a. plots resistivity vs. the voltage-drop on the WSM, at different V_G . Eq. 2 refers to the data shown in Fig. 4b. A full derivation of Eq. 2 is reported in the *Supplementary*. We have added a new figure in the supplementary to address the point raised by the reviewer.

Changes: We added in the *Supplementary* a new figure (Fig. S3b), which is complementary to Fig. 3c and shows the linear dependence of the modulation on V_G , for the different field offsets.

REVIEWER COMMENTS

Reviewer #1 (Remarks to the Author):

The authors have addressed all of my comments and I recommend publication in Nature Communications.

Reviewer #3 (Remarks to the Author):

The reviewer still has some concerns about the transistor operation of the device, and it would be good if the authors could clarify the two points:

1. The paper should include the transistor voltage and current gain, if they are much below unity this should be commented upon, since then the transistor does not show any gain. Without any small signal gain calling the device a transistor can be a bit confusing.
2. Please clarify how the transconductance is obtained – $4a$ seems to indicate g_m varying with v_{gs} . I would recommend to include output and transfer characteristics, as well as $g_m(v_{gs})$.

RESPONSE TO REFEREES

Reviewer #3 (Remarks to the Author):

1. *The paper should include the transistor voltage and current gain, if they are much below unity this should be commented upon, since then the transistor does not show any gain. Without any small signal gain calling the device a transistor can be a bit confusing.*

Reply: Concerning current gain, we reported in the supplementary material the expression one would use to calculate it. However, as we noted, Z_G is in principle zero in DC, therefore the current gain is not well-defined in the ideal case, but depends on the real R_G , which can be tuned arbitrarily. The voltage gain can be calculated in a similar manner, using the WSM resistance instead of Z_G . The value we would obtain in this case is below the unit, due to the low resistance of the WSM. The use of the word transistor has been made to highlight the potential operation of such class of the device, even if the demonstrator we realized does not show a significant gain. What we highlighted is the transconductance gain, meaning the device is able to effectively transduce a voltage variation into a current variation. More importantly, we have shown in our prior simulation work that small signal gain and overall excellent high-frequency properties could be achieved given access to thin-film epitaxy and other innovations outlined above. For these reasons, we feel confident in calling this device a transistor.

Changes: Modified sentence (lines 60-63) to better highlight the amplifying properties of the device. Added a sentence (lines 49-54) in the supplementary to discuss about gain metrics.

2. *Please clarify how the transconductance is obtained – 4a seems to indicate gm varying with vgs. I would recommend to include output and transfer characteristics, as well as gm(vgs).*

Reply: We thank the reviewer for this comment. The transconductance is evaluated from the dataset showed in Fig. 4a, even though the purpose of the figure is mainly to comment about possible self-heating of the WSM crystallite. Therefore, we added in the supplementary a new figure (Fig. S5), which reports the transfer and output characteristic. The transfer characteristic (Fig. S5a) has been referenced to the value at $V_G = 0$ V to compensate for the voltage separation in the characteristic. From that plot (ΔV_{WSM} vs. V_G) it is possible to evaluate the transconductance by taking the derivative of the two and dividing by the WSM resistance. The output characteristic (Fig. S5b) has been plotted at the bias point for visualization purposes, since the modulation is small compared to the voltage range that we selected. From it, it is possible to read the current variation induced by the gate voltage. Lastly, concerning g_m as a function of V_{GS} , we decided not to plot it for the simple reason that the gate voltage dependence is almost negligible, given that K_G in Eq. 2 is small compared to the unity.

Changes: Added two new figures in the supplementary (Fig. S5a and b). Added a new sentence in the main text (lines 235-236) to better explain how the transconductance is evaluated.

REVIEWERS' COMMENTS

Reviewer #3 (Remarks to the Author):

1) Some (minor?) ambiguities - for a device with zero DC input resistance, the current gain should be well defined ($A_i = di_{out}/di_{in}$), whereas the transconductance should depend on the DC gate biasing resistance. ($g_m = di_{out}/dv_{in}$)? The opposite is claimed in the paper.

2) From the S5B plot, a μS ($= (di_{WSM}/dV_g) \sim 2\mu A/2V$) transconductance instead of a mS transconductance would be reasonable?

RESPONSE TO REFEREES

Reviewer #3 (Remarks to the Author):

1. *Some (minor?) ambiguities - for a device with zero DC input resistance, the current gain should be well defined ($A_i = di_{out}/di_{in}$), whereas the transconductance should depend on the DC gate biasing resistance. ($gm = di_{out}/dv_{in}$)? The opposite is claimed in the paper.*

Reply: We thank the reviewer for this comment. The current gain can be defined as $A_I = dI_{out}/dI_{in}$, however, since in our case the gate signal is driven by a voltage, the relationship between V and I is not well-defined due to the zero input resistance.

2. *From the S5B plot, a μS ($= (di_{WSM}/dVg) \sim 2\mu A/2V$) transconductance instead of a mS transconductance would be reasonable?*

Reply: We thank the reviewer for this comment. The reported value of transconductance ($\sim 2\mu A/2V$) would be the nominal transconductance of the device, without taking in consideration the contact resistance of the gate. Because of the way the signal is transferred, in the main text we refer to the reported transconductance as extrinsic transconductance, which considers the actual voltage drop on the gate contacts, measured in a 4-point configuration.

Changes: Added a sentence in the caption of Fig. 4b (last sentence) to further clarify how g_m is computed.